# Predicting Sarcopenia in Peritoneal Dialysis Patients: A Multimodal Ultrasound-Based Logistic Regression Analysis and Nomogram Model

**DOI:** 10.3390/diagnostics15212685

**Published:** 2025-10-23

**Authors:** Shengqiao Wang, Xiuyun Lu, Juan Chen, Xinliang Xu, Jun Jiang, Yi Dong

**Affiliations:** Department of Ultrasound, Xinhua Hospital Affiliated to Shanghai JiaoTong University, School of Medicine, 1665th Kongjiang Road, Shanghai 200092, China; wang_shengqiao520@126.com (S.W.); lxy_dolly@126.com (X.L.); rainy1208@126.com (J.C.); xuxinliang1126@foxmail.com (X.X.); tenine@163.com (J.J.)

**Keywords:** peritoneal dialysis (PD), sarcopenia, logistic regression, nomogram, attenuation coefficient (Atten Coe), prediction model

## Abstract

**Objective:** This study aimed to evaluate the diagnostic value of logistic regression and nomogram models based on multimodal ultrasound in predicting sarcopenia in patients with peritoneal dialysis (PD). **Methods:** A total of 178 patients with PD admitted to our nephrology department between June 2024 and April 2025 were enrolled. According to the 2019 Asian Working Group for Sarcopenia (AWGS) diagnostic criteria, patients were categorized into sarcopenia and non-sarcopenia groups. Ultrasound examinations were used to measure the muscle thickness (MT), pinna angle (PA), fascicle length (FL), attenuation coefficient (Atten Coe), and echo intensity (EI) of the right gastrocnemius medial head. The clinical characteristics of the groups were compared using the Mann–Whitney U test. Binary logistic regression was used to identify sarcopenia risk factors to construct clinical prediction models and nomograms. Receiver operating characteristic (ROC) curves were used to assess the model accuracy and stability. **Results:** The sarcopenia group exhibited significantly lower MT, PA, and FL, but higher Atten Coe and EI than the non-sarcopenia group (all *p* < 0.05). A multimodal ultrasound logistic regression model was developed using machine learning—Logit(P) = −7.29 − 1.18 × MT − 0.074 × PA + 0.48 × FL + 0.52 × Atten Coe + 0.13 × EI (*p* < 0.05)—achieving an F1-score of 0.785. The area under the ROC curve (ROC-AUC) was 0.902, with an optimal cut-off value of 0.45 (sensitivity 77.3%, specificity 56.7%). Nomogram consistency analysis showed no statistical difference between the ultrasound diagnosis and the appendicular skeletal muscle index (ASMI) measured by bioelectrical impedance analysis (BIA) (Z = 0.415, *p* > 0.05). **Conclusions:** The multimodal ultrasound-based prediction model effectively assists clinicians in identifying patients with PD at a high risk of sarcopenia, enabling early intervention to improve clinical outcomes.

## 1. Introduction

Sarcopenia is primarily characterized by decreased muscle strength, reduced muscle mass, and diminished physical function, and is associated with adverse outcomes such as falls, frailty, and mortality [1]. Sarcopenia is classified into primary sarcopenia, which is related to aging, and secondary sarcopenia, which is linked to prolonged immobilization, severe malnutrition, and chronic kidney disease, among other factors [2,3]. Several international organizations and expert working groups have published diagnostic criteria for sarcopenia [2], including the standards proposed by European Working Group on Sarcopenia in Older People (EWGSOP) and Asian Working Group for Sarcopenia (AWGS).

Peritoneal dialysis (PD) is the most common replacement therapy for end-stage renal disease (ESRD), and adverse factors such as nutrient loss or inadequate uptake during prolonged dialysis, hormonal imbalances, electrolyte disturbances, and chronic inflammation can accelerate the process of protein-energy depletion, leading to increased muscle and fat loss and a high risk of sarcopenia [4,5,6]. Abro et al. [7] found that the incidence of sarcopenia in patients with chronic kidney disease PD was between 11.0% and 15.5%. Kamijo et al. [8] found that the incidence of sarcopenia was 10.9% in 119 patients with PD.

The measurement of muscle mass is an important step in the diagnosis of sarcopenia. Commonly used techniques for this purpose include bioelectrical impedance analysis (BIA), dual-energy X-ray absorptiometry (DXA), computed tomography (CT), magnetic resonance imaging (MRI), and ultrasound (US) [9].

However, the accuracy of BIA and DXA may be influenced by a patient’s hydration status and adiposity [10,11]. MRI is more expensive and has some limitations, such as contraindications to metal implants and a lack of diagnostic criteria. CT is considered to be the gold-standard method for measuring muscle mass, but it is not suitable for widespread screening for sarcopenia due to radiation exposure and high cost [12,13]. Therefore, it is particularly important to identify a validated and suitable assessment tool for clinical use.

Ultrasound, as a noninvasive imaging method, can be used to assess muscle structure, stiffness, and microvascular perfusion using B-mode ultrasound, elastography, and high-resolution microvascular imaging which have the advantages of real-time, convenient, and easy bedside operation. The use of ultrasound to measure muscle thickness (MT), pinna angle (PA), and fascicle length (FL) to estimate muscle mass has been reported [14,15,16]. B-mode ultrasound imaging can assess muscle structure and echo intensity (EI) [17,18], while the Raw vision attenuation (RVA) technique can quantitatively measure muscle attenuation values to reflect muscle hardness [19]. RVA is a “frequency-slope” algorithm that calculates the acoustic attenuation coefficient (dB·cm^−1^·MHz^−1^) from raw echo signals within the 8–14 MHz range. It is independent of grayscale values and can directly reflect intramuscular fat/fiber content [20,21]. Due to its advantages of being non-invasive, free from fluid interference, and requiring a short operation time, RVA has been clinically applied in patients with ascites and edema, and those with metal implants or pacemaker contraindications for BIA [10].

Many previous studies have reported predictive models for the occurrence of sarcopenia in elderly and diabetic patients; however, relatively few studies have focused on predictive models for sarcopenia in patients undergoing PD [22,23]. The previous study relied solely on routine BMI, handgrip strength, and mid-arm muscle circumference (MAMC) to provide a simple tool for sarcopenia. The present article integrates multimodal ultrasound parameters into a nomogram, offering an imaging-based alternative for centers without access to BIA or DXA and thereby enhancing diagnostic accuracy. The nomogram, which has gained widespread application in disease diagnosis and prediction in recent years, can integrate multiple variables and visually display their relative importance through a scoring system [24].

Therefore, our research design is as follows: After participant recruitment and on-site assessment of muscle mass and strength (BIA and grip test), all subjects underwent blinded ultrasound examination of the right medial gastrocnemius. Parameters with significant between-group differences were entered into a multivariable logistic model, internally validated by bootstrap, and translated into a nomogram. The final section compares model performance with BIA-based diagnosis and discusses deployment issues, thereby providing a scientific reference for the early screening of sarcopenia in this patient population with PD.

## 2. Materials and Methods

### 2.1. Patients

Between June 2024 and April 2025, a total of 178 patients undergoing peritoneal dialysis in the nephrology department were included. According to Asian Working Group for Sarcopenia: 2019 Consensus Update on Sarcopenia Diagnosis and Treatment [25]. The inclusion criteria were as follows: (1) regular peritoneal dialysis for ≥3 months; (2) age ≥ 18 years, aware of the study content, and voluntarily participated. The exclusion criteria were as follows: (1) physical intolerance to bioelectrical impedance analysis (BIA) or grip strength/gait speed tests; (2) comorbidities such as malignant tumors, tuberculosis, liver disease, or peptic ulcers; (3) acute infections or cardiovascular events within the last 6 months; (4) pre-existing primary muscle-related diseases. The patients were divided into sarcopenia (60 patients) and non-sarcopenia (118 patients) groups. The sarcopenia/non-sarcopenia classification was not based on prior clinical diagnoses. It was time-separated from image acquisition, and was performed under explicit blinding.

### 2.2. Demography Characteristics

Basic information about the patients was recorded, including age, sex, height, weight, and body mass index (BMI). Laboratory indicators, including albumin, calcium, phosphorus, creatinine, and glomerular filtration rate were collected. Nutritional status was assessed using the Subjective Global Assessment (SGA).

### 2.3. Muscle Mass and Muscle Strength Estimation

Muscle mass and strength were assessed according to the 2019 AWGS standard [25]. A bioelectrical impedance analyzer (InBody s10, Shanghai, China) was used to measure body composition after dialysis was completed. The measurements included BMI and appendicular skeletal muscle mass index (ASMI). Muscle strength was assessed using an electronic handgrip strength meter (CAMRY EH101, Guangzhou, China). The patient’s elbow was extended, and they were in a standing position, with both upper limbs naturally sagging on their sides. Both hands were measured twice, with an interval of 3 min between each measurement, and the maximum value was recorded.

### 2.4. Ultrasound Examination

All ultrasound examinations were performed using an eSonic eHertz ultrasound diagnostic instrument (Beijing eSonic Medical Technology Co., Beijing, China) with an L16-4Ep linear array probe (4–16 MHz). The linear array probe precisely matches the anatomical depth of the calf gastrocnemius and can clearly display fine structures such as muscle fiber bundles and fascia in high-frequency mode. The same probe incorporates elastography and RVA modules, allowing multimodal acquisition without transducer change, thus shortening examination time and reducing inter-operator variability [26].

The patients were placed in the prone position, with the ankle joint placed outside the examination bed. The probe was placed on top of the skin with minimal load to ensure that no external pressure could affect the tests. The musculoskeletal mode was selected, and the right medial head of the gastrocnemius was viewed in the upper middle 1/3 of the line from the medial tibial condyle to the highest point of the medial malleolus [17]. The probe was parallel to the long axis of the gastrocnemius muscle, and MT, PA, and FL were measured in a relaxed state (Figure 1A–C). The two-dimensional cross-sectional images were imported into Adobe Photoshop 23, and the EI was measured (Figure 1D). Because pixel intensity is sensitive to time gain compensation (TGC), a dedicated “Muscle-Sarcopenia” preset was created and locked before patient enrolment. Total gain was fixed at 72 dB; TGC sliders were set to the factory linear curve. Dynamic range (60 dB), frequency (12 MHz), depth (3 cm) and single focal zone (1.5 cm) were identical for every scan. EI was quantified by placing a 10 × 5 mm ROI between the aponeuroses; the mean gray value (0–255) was used for analysis.

On the longitudinal section, a novel RVA quantitative evaluation technique was per formed using an ultrasound system. The rectangular region of interest (ROI) was positioned at the image center, where an automated system tool for RVA quantification (4 mm diameter) was placed at the center of the ROI. The attenuation coefficient (Atten Coe) was measured on a scale ranging from 0.10 dB/cm/MHz (blue) to 1.80 dB/cm/MHz (red) (Figure 1E). All subjects were examined by the same sonographer. The above acquisitions were measured three times consecutively, and the average value was obtained. The individual values had to lie within ±5% of their mean, otherwise the set was repeated. The final value entered into the database was the arithmetic mean of the three accepted measurements, with an intra-set coefficient of variation (CV) consistently <4%, the values obtained all satisfied the homogeneity of variance.

### 2.5. Diagnostic Criteria for Malnutrition

Patients with PD were evaluated for their nutritional status using the Patient-Generated Subjective Global Assessment (PG-SGA). A final score of ≤1 indicates good nutritional status, while a score of >1 indicates that the patient may have malnutrition.

### 2.6. Diagnostic Criteria for Sarcopenia

The diagnosis of sarcopenia was based on the 2019 AWGS [25]: (1) low muscle mass: ASMI measured by BIA < 7.0 kg/m^2^ in males and <5.7 kg/m^2^ in females; (2) low muscle strength: handgrip strength < 28 kg in males and <18 kg in females; and (3) physical mobility: low gait speed cut-off value < 1.0 m/s. In this study, the sarcopenia group met criteria (1) and (2).

### 2.7. Prediction Model Construction

The modelling cohort consisted of 178 participants with complete outcome data. Missing values (accounting for 0.34% of the total data) were imputed five times using chained equations, and pooled estimates are reported [27]. Variables with a variance inflation factor (VIF) ≥ 5 or a univariable *p*-value ≥ 0.10 in the initial screening were excluded. A multivariable logistic regression model was then constructed using a backward elimination procedure, with a significance level of α = 0.05 for variable retention [28]. The model’s performance was internally validated using bootstrap resampling with replicates to assess its stability and optimism [29]. The final model was presented as a nomogram by linearly converting each regression coefficient (β) to a 0–100 point scale, facilitating clinical interpretation and use [30]. The discriminative ability of the model was evaluated using the area under the receiver operating characteristic curve (ROC-AUC). Additionally, the F1 score was calculated to reflect the balance between sensitivity and specificity. The model was considered to have optimal performance when the predicted probability score approached 1, indicating a high likelihood of sarcopenia.

### 2.8. Statistical Analyses

The data were processed using SPSS 26.0 statistical software and R language (version 4.3.1). Normally distributed data were expressed as mean ± standard deviation, and independent sample *t*-test was used for intergroup comparison. Non-normally distributed measurement data were expressed as median and interquartile range, and the independent sample Mann–Whitney U test was used for comparison between the two groups. Count data were expressed as numbers (%), and the χ^2^ test was used for intergroup comparisons.

## 3. Results

### 3.1. General Information of Peritoneal Dialysis (PD) Patients

A total of 178 patients with PD were included in our study. 60 (33.70%) patients were included in the sarcopenia group, and 117 (66.30%) were included in the non-sarcopenia group. By comparing the general conditions of the two groups of patients, there were statistically significant differences in BMI, diabetes, malnutrition status, albumin, calcium, phosphate, handgrip strength, and ASMI (*p* < 0.05). There were no statistical differences in age, gender, dialysis duration, hypertension, creatinine levels, and Glomerular Filtration Rate (GFR) (*p* > 0.05) (Table 1).

### 3.2. Ultrasound Parameter

The parameters measured by ultrasound in the relaxed state of each group are presented in Figure 2. Compared with the non-sarcopenia group, the MT, PA, and FL of the sarcopenia group were lower than those of the non-sarcopenia group, the Atten Coe and EI was higher than that of the non-sarcopenia group.

### 3.3. Multivariate Logistic Regression Analysis of Sarcopenia in Peritoneal Dialysis (PD)

The occurrence of sarcopenia was associated with multiple factors (*p* < 0.05). These indicators were included in the multivariate logistic regression analysis. The results showed that BMI, nutritional status, MT, PA, FL, Atten Coe, and EI were influencing factors for sarcopenia in PD patients (Table 2).

### 3.4. Correlation Analysis Between Ultrasound Indicators and Clinical Diagnostic Parameters of Sarcopenia

The analysis of correlations between MT, PA, FL, Atten Coe, EI, and clinical diagnostic parameters for sarcopenia are shown in Figure 3. MT exhibited positive correlations with handgrip strength (r = 0.93, *p* < 0.05) and ASMI (r = 0.98, *p* < 0.05), while Atten Coe exhibited negative correlations with handgrip strength (r = −0.73, *p* < 0.05) and ASMI (r = −0.76, *p* < 0.05). EI exhibited negative correlations with handgrip strength (r = −0.93, *p* < 0.05) and ASMI (r = −0.91, *p* < 0.05)

### 3.5. Constructing a Model for Sarcopenia in Peritoneal Dialysis (PD) Patients

These results indicate that MT, PA, FL, Atten Coe, and EI were correlated with sarcopenia. We used the ROC curve to explore whether the variables could act as diagnostic markers to identify sarcopenia in patients with PD. A Logistic regression model: Logit(*P*) = −7.29 − 1.18 × MT − 0.074 × PA + 0.48 × FL + 0.52 × Atten Coe + 0.13 × EI (*p* < 0.05). The F1 score was 0.785, and the ROC-AUC reached 0.902, The best cutoff value of the combined ultrasound indices for the diagnosis of sarcopenia was 0.45, with a sensitivity of 77.3% and specificity of 56.7%.

### 3.6. Construction and Evaluation of a Nomogram Prediction Model

A nomogram prediction model was constructed based on the logistic regression model (Figure 4). Each variable is first converted into points using the top scale (0–100). Add the points; the total-points scale gives the probability of sarcopenia (0–1). The ROC-AUC of the nomogram model for predicting sarcopenia was 0.902 (95% CI: 0.877–0.924), with a sensitivity of 77.3% and a specificity of 56.7%. The ROC-AUC of the model was larger than any independent variable. Consistency analysis between the combined ultrasound indicators and ASMI measurements obtained using the BIA method showed no statistically significant difference in the diagnostic results of sarcopenia between the combined ultrasound indicators and ASMI (Z = 0.415, *p* > 0.05) (Figure 5).

## 4. Discussion

This study focused on developing a predictive model for sarcopenia in patients undergoing peritoneal dialysis (PD) using multimodal ultrasound parameters combined with logistic regression analysis and nomogram modeling. Sarcopenia is common in PD patients and is associated with adverse outcomes such as physical dysfunction, reduced quality of life, and higher mortality risk [31,32]. Early detection of high-risk patients is crucial for timely clinical interventions to slow disease progression [33]. Compared to traditional methods, ultrasound offers non-invasive, convenient, highly reproducible, and bedside assessment advantages, making it suitable for widespread clinical applications. Both ultrasound and bioelectrical impedance analysis (BIA) have short exam durations, with a time difference of <2 min, making them suitable for routine PD outpatient visits [34]. In terms of reliability, ultrasound is generally superior to BIA: it is more stable especially in special physiological conditions (e.g., fluid overload or ascites) and has no restrictions on metal implants or pacemakers [35,36]. For routine detection of fat, muscle ratio, or muscle mass, BIA is far cheaper than ultrasound. This is due to BIA’s relatively simple technology, which requires no complex consumables or high-skill operations [37].

In our study, the sarcopenia group exhibited significantly lower muscle thickness (MT), pinna angle (PA), and fiber length (FL) than the non-sarcopenia group, whereas the attenuation coefficient (Atten Coe) and echo intensity (EI) were significantly higher. These findings suggest significant structural and textural alterations in the muscle tissue of patients with sarcopenia. The literature indicates that muscle loss in sarcopenia is primarily attributed to the transformation from type II to type I fibers, a reduction in muscle fiber quantity, a decrease in muscle volume, and disorganized muscle fiber arrangement, all of which directly affect muscle function [38]. Sarcopenia is characterized by diminished muscle strength, reduced muscle volume, and decreased cross-sectional area of the muscle fibers [39].

MT directly reflects muscle mass, with greater thickness indicating a higher muscle mass. PA is closely related to the number of contracted muscle fibers and is commonly used to evaluate the relationship between muscle contraction function and strength. A larger pinna angle corresponds to more contracted muscle fibers and a stronger force-generating capacity. FL is proportional to the speed and range of the muscle contraction. Decreased muscle mass results in reduced muscle strength and stiffness in older adults [40,41]. Our findings are consistent with these reports regarding muscle mass.

The Atten Coe reflects tissue density and texture, with higher values typically indicating increased intramuscular fat and fibrous tissue, especially in patients with sarcopenia [42]. Muscle echogenicity, an age-related change linked to muscle atrophy and fatty infiltration [43], refers to the grayscale value of muscle tissue in ultrasound images. It reflects tissue homogeneity and the content of non-contractile components, such as fat and fibrous tissue. Higher echogenicity usually signifies more non-contractile components, which are closely associated with decreased muscle mass and impaired function [44]. The Atten Coe and echogenicity are crucial parameters for ultrasound muscle assessment. They reflect the structural and textural changes in muscle tissue and are strongly correlated with the decline in muscle mass and function. These parameters are valuable for the early diagnosis and monitoring of sarcopenia in older adults.

Our study demonstrated that the sarcopenia group had a lower BMI and a higher prevalence of malnutrition than the non-sarcopenia group. Multivariate logistic regression analysis identified factors influencing sarcopenia in PD patients, including BMI, nutritional status, MT, PA, FL, Atten Coe and EI. Lower BMI and malnutrition were positively correlated with sarcopenia in this study. This study highlights the complex relationship between BMI and sarcopenia, where a lower BMI is associated with a higher risk of sarcopenia, whereas a higher BMI has a protective effect [45,46]. Malnutrition may reduce muscle mass and increase the risk of sarcopenia [37]. Therefore, both BMI and nutritional status should be comprehensively evaluated when assessing sarcopenia, and these factors should be prioritized in clinical practice for early screening and intervention in high-risk patients.

Our study found that multimodal ultrasound characteristic parameters were correlated with the clinical diagnostic parameters of sarcopenia. MT was positively correlated with handgrip strength and appendicular skeletal muscle index (ASMI), whereas the Atten Coe and EI were negatively correlated with these clinical parameters. These findings align with previous studies showing positive correlations between muscle thickness and handgrip strength/ASMI and negative correlations between echo intensity and handgrip strength/ASMI [47,48,49].

ASMI is a widely accepted method for measuring muscle mass, and grip strength is an indicator of overall muscle strength. The Asian Working Group for Sarcopenia (AWGS) consensus suggests that a smaller ASMI indicates a higher risk of sarcopenia. Hida et al. [50] found a significant correlation between ASMI and ultrasound measurements in their study of elderly muscle measurements. Our research is consistent with previous studies [51,52] showing a positive correlation between ASMI and MT, PA, and FL. Grip strength has been suggested as a significant predictor of sarcopenia and shows a significant correlation with ASMI [53,54].

The multimodal ultrasound logistic regression model developed in this study exhibited outstanding predictive performance, with an F1 score of 0.785 and an ROC-AUC value of 0.902, demonstrating high accuracy in predicting sarcopenia among patients undergoing PD. At an optimal cutoff value of 0.45, the model achieved 77.3% sensitivity and 56.7% specificity. Although the specificity was relatively modest, the model effectively minimized false negatives, ensuring the timely identification of potential sarcopenia cases. Calibration analysis of the nomogram prediction model showed no statistically significant difference between ultrasound diagnosis and ASMI measured by BIA, further validating the reliability of ultrasound parameters for sarcopenia assessment. This indicates that clinicians can utilize this model in conjunction with ultrasound findings to rapidly and accurately evaluate the risk of sarcopenia in patients with PD, thereby providing valuable guidance for personalized treatment strategies. In terms of system dependency, though the five ultrasound features are physical quantities, their numerical outputs are vendor-specific, meaning different ultrasound systems may alter the model. Ultrasound cannot replace hand grip dynamometry measurements; however, for patients with active ascites, metal implants, or obesity, ultrasound can replace BIA for muscle mass assessment. In summary, this technique is designed as an interoperable frontline tool that integrates with, rather than replaces, existing AWGS pathways, and calibrating local hardware is mandatory before its clinical rollout.

This study had several limitations. First, the relatively small sample size and single-center design may limit the generalizability and extrapolation of the results. Future research should involve multicenter studies with larger sample sizes to further validate the model’s stability and applicability. Second, although the same sonographer performed all ultrasound parameter measurements and took multiple measurements to calculate averages to reduce errors, there may still be some subjectivity and measurement error. Future studies should explore more advanced ultrasound imaging technologies and automated measurement tools to improve measurement accuracy and objectivity. Additionally, this study focused solely on predicting sarcopenia without evaluating the effectiveness of the intervention measures. Subsequent research could investigate ultrasound-monitored sarcopenia intervention approaches, such as nutritional support and exercise rehabilitation, to improve the muscle status and clinical prognosis of patients undergoing peritoneal dialysis.

## 5. Conclusions

In summary, this study introduces a new method and tool for early sarcopenia prediction in patients with peritoneal dialysis (PD) through a multimodal ultrasound-based logistic regression and nomogram model, showing considerable potential for clinical application. Future studies should focus on further optimizing the model, expanding the research scope, and exploring effective interventions to better address the issue of sarcopenia in this patient group.

## Figures and Tables

**Figure 1 diagnostics-15-02685-f001:**
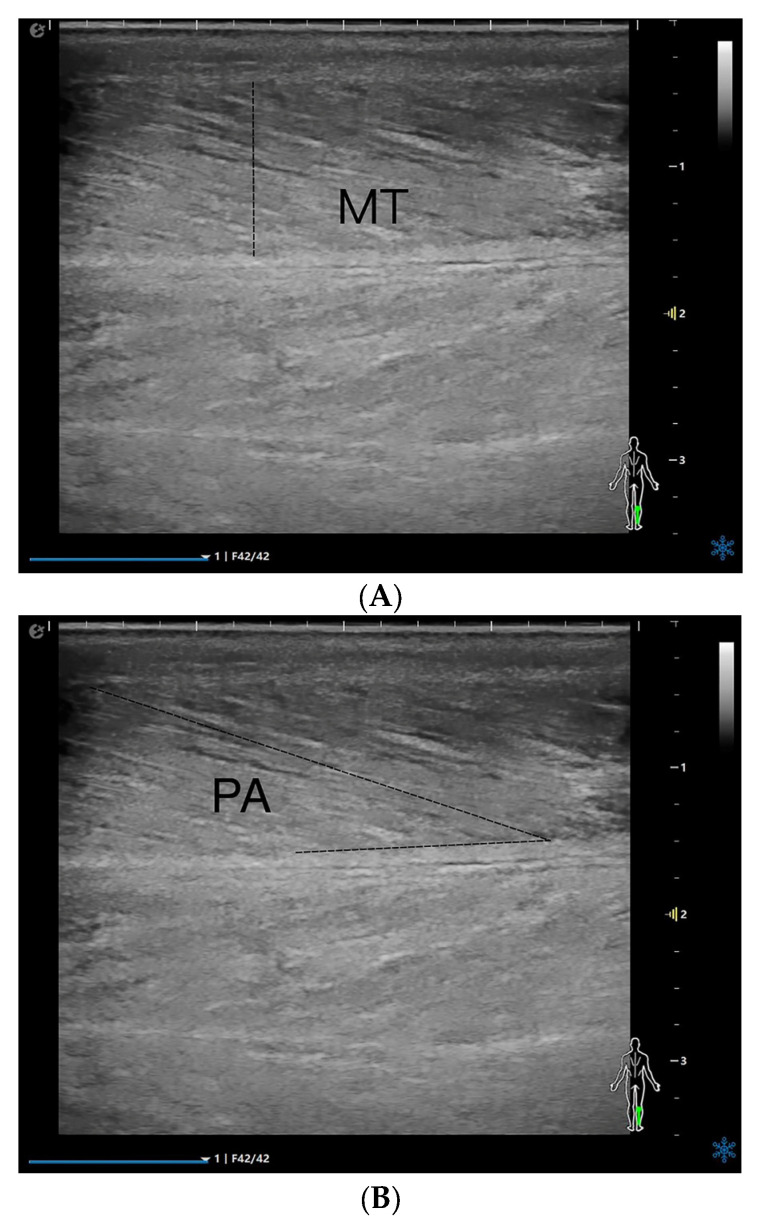
MT (**A**), PA (**B**), FL (**C**), EI (**D**), and Atten Coe (**E**) of the medial head of the right gastrocnemius muscle in the relaxed state. MT = muscle thickness (mm); PA = pinna angle (°); FL = fascicle length (mm); Atten Coe = attenuation coefficient (dB/cm/MHz); EI = echo intensity.

**Figure 2 diagnostics-15-02685-f002:**
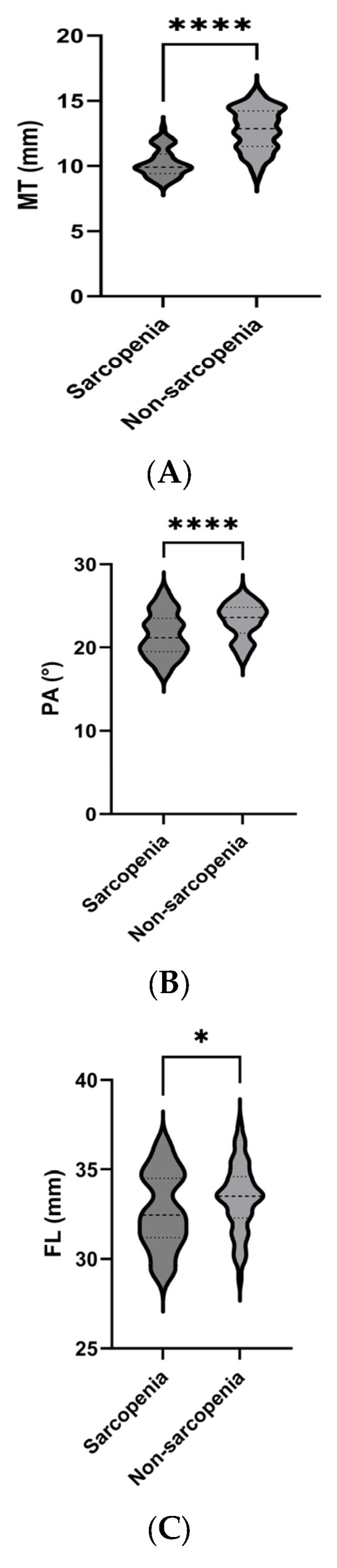
Comparison of ultrasound parameters between the sarcopenia and non-sarcopenia groups. MT (**A**), PA (**B**), FL (**C**), Atten Coe (**D**), and EI (**E**). * *p* < 0.05, **** *p* < 0.001. MT = muscle thickness (mm); PA = pinna angle (°); FL = fascicle length (mm); Atten Coe = attenuation coefficient (dB/cm/MHz); EI = echo intensity.

**Figure 3 diagnostics-15-02685-f003:**
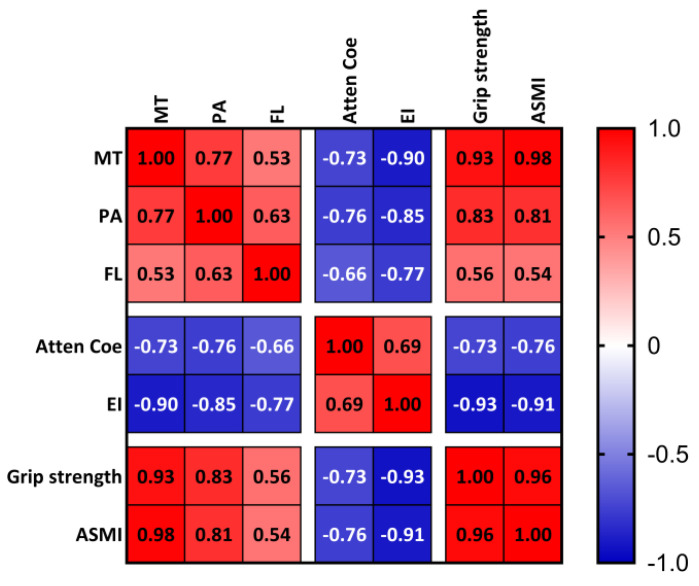
Correlation analysis between ultrasound indicators and clinical diagnostic parameters of sarcopenia. MT = muscle thickness (mm); PA = pinna angle (°); FL = fascicle length (mm); Atten Coe = attenuation coefficient (dB/cm/MHz); EI = echo intensity.

**Figure 4 diagnostics-15-02685-f004:**
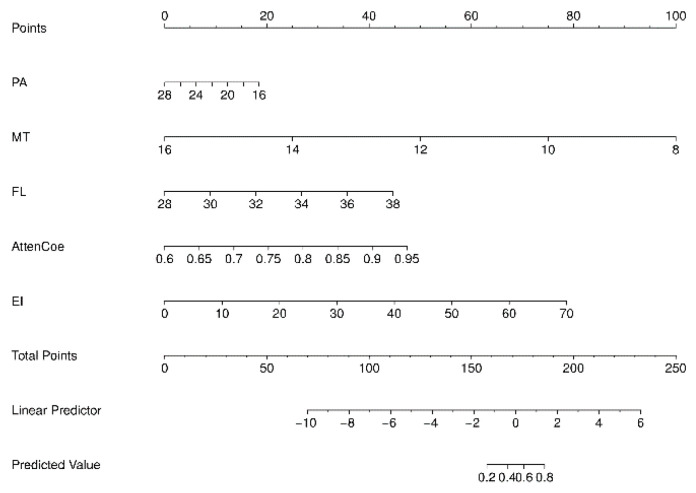
Nomogram prediction model. MT = muscle thickness (mm); PA = pinna angle (°); FL = fascicle length (mm); Atten Coe = attenuation coefficient (dB/cm/MHz); EI = echo intensity.

**Figure 5 diagnostics-15-02685-f005:**
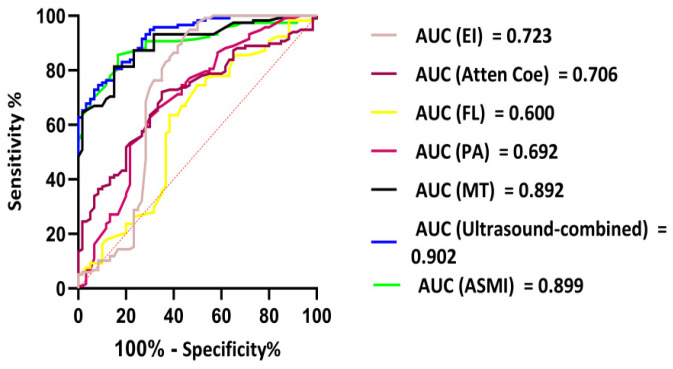
ROC analyses results of ultrasound-combined index values in predicting sarcopenia.

**Table 1 diagnostics-15-02685-t001:** Comparison of general information between sarcopenia and non-sarcopenia groups.

Characteristics	Sarcopenia (*n* = 60)	Non-Sarcopenia (*n* = 118)	*p*
Age, years	70.19 ± 7.09	68.88 ± 8.99	0.063
Male, n (%)	35 (58.33)	62 (52.54)	0.463
Dialysis duration, years	5 (3, 8)	6 (2, 9)	0.982
BMI, kg/m^2^	23.35 ± 2.36	24.08 ± 1.36	0.016
Hypertension, n (%)	54 (90.00)	108 (91.52)	0.549
Diabetes, n (%)	47 (78.33)	72 (61.22)	0.002
Malnutrition, n (%)	42(70.00)	56 (47.45)	0.004
Albumin, g/L	30.85 ± 2.75	31.22 ± 2.73	0.033
Calcium, mmol/L	2.16 ± 0.13	2.19 ± 0.18	0.038
Phosphate, mmol/L	1.45 ± 0.26	1.50 ± 0.37	0.022
Creatinine, umol/L	846.22 ± 230.89	832.61 ± 222.92	0.704
GFR, mL/min/1.73 m^2^	5.09 ± 1.26	5.51 ± 2.22	0.176
Handgrip strength, kg	18.45 ± 5.04	28.53 ± 4.50	<0.001
ASMI, kg/m^2^	5.59 ± 0.65	7.41 ± 0.96	<0.001

BMI = Body mass index; GFR = Glomerular Filtration Rate; ASMI = Appendicular skeletal muscle mass index.

**Table 2 diagnostics-15-02685-t002:** Multivariate logistic regression analysis.

	β	SE	Wald	*p*	OR (95%CI)
BMI	−0.172	0.086	4.058	0.044	0.842 (0.712, 0.995)
Nutritional	0.949	0.337	7.947	0.005	2.583 (1.335, 4.998)
MT	−1.227	0.228	28.854	<0.01	0.293 (0.187, 0.459)
PA	−0.151	0.110	1.897	0.036	0.829 (0.473, 1.453)
FL	0.438	0.161	7.405	0.024	0.666 (0.306, 1.140)
Atten Coe	13.283	5.386	6.081	0.018	0.742 (0.535, 1.108)
EI	0.110	7.304	4.402	0.002	0.735 (0.506, 1.035)
Constant	−13.235	7.806	2.875	0.09	

BMI = body mass index (kg/m^2)^; MT = muscle thickness (mm); PA = pinna angle (**°**); FL = fascicle length (mm); Atten Coe = attenuation coefficient (dB/cm/MHz); EI = echo intensity. Nutritional status: Binary variable (0 = well-nourished, 1 = malnourished). Data are presented as regression coefficient (β), standard error (SE), odds ratio (OR) and 95% confidence interval (CI).

## Data Availability

The original contributions presented in this study are included in the article. Further inquiries can be directed to the corresponding author.

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
