# Peer review of "Predicting Sarcopenia in Peritoneal Dialysis Patients: A Multimodal Ultrasound-Based Logistic Regression Analysis and Nomogram Model"

_diagnostics, 2025, doi:10.3390/diagnostics15212685_

Round 1

Reviewer 1 Report

Comments and Suggestions for Authors

Dear authors 

I read with interest Your paper regarding multimodal ultrasound approach in sarcopenia.

From my point of view there are some problems which need to be resolvced:

The raw vision attenuation is "new" but is lacking confirmation in other studies, at the moment it seems to be more like a software application in your ultrasound machine. Some more specifications and literature backgfround should be given.

To measure "echo intensity" you need to export your foto in order to apply Adobe Photoshop 23 for the calculation. I think this approach is not very practicable as a bedside / point of care approach.

Leaving away these two above mentioned "special" measurement you found a deterioration of the sensitivity and specificity of ultrasound in detecting/predicting sarcopenia? How would be influenced the logistic regression model? 

As the ultrasonographer calculated the mean of three measurements you should mention the variance of the values obtained.

The figure 2 is without scaling, thus needing modifications. It might be even interesting to present the ultrasound values in a separate table differentiating in sarcopenia and non-sarcopenia group (as you already did in table 1).

The clinical applicability of the "nomogram" is not clear for the reader, it seems to be a result of a statistical computer program. The units of the variables are not specified.

Reviewer 2 Report

Comments and Suggestions for Authors

The authors present a clinical study of ultrasonic measures for development of a predictive model for sarcopenia. The topic should be of interest to the journal's readers, but substantially more detail is needed on the construction of the predictive model and how it would ultimately be deployed. 

Line specific comments and questions:

L56-57:This is a list of methods, but not a sentence to tie them together

L67: Find an objective term to replace 'superb'

L74: 'Relatively few studies have focused on predictive models for sarcopenia in patients undergoing PD'. The authors should present and cite these studies, and explain how the present study differs.

L82: Before starting the methods section it would be helpful to briefly describe what will follow.

L85: Can the authors provide justifications for their inclusion and exclusion criteria?

L92: Was the division between sarcopenia/non-sarcopenia based on prior diagnoses? Did this division occur before the ultrasound testing, and if so, was there a potential for biasing?

L103: If the ultrasound assessment was carried out on the calf muscle, why was strength assessed using hand grip measurements? Is it possible that calf strength and hand strength would be dissimilar depending on activity?

L109: Can the authors provide a justification for the procedural choices in the ultrasound exam? 

L119: Please explain the EI measurement in greater detail, as this quantity can vary with US system settings, particularly time gain compensation, unless these were purposefully kept constant for all exams. 

L151: Much more detail is needed here about the models. 

Figure 2: thickness, angle, attenuation and intensity all have different units of measure. They should therefore be put on separate plots with correct units displayed.

The measures in Table 2 need to be formally defined. 'Nutritional' is incorrectly spelled.

L232: How do the practicalities of ultrasound and bioimpedance measurements compare, in terms of time/cost/reliability of the exams?

L296: Would the model change with the use of a different ultrasound system? If so, how would this technique best be deployed? More generally, how would the ultrasound diagnostics be used - in addition to or instead of other existing methods?

Round 2

Reviewer 2 Report

Comments and Suggestions for Authors

The authors made several improvements. Some further refinements are needed:

1. The author responses to comments 6, 9 and 14 should be included in the manuscript.

2. The model description is still not as clear as it could be, and a few references on design of models of this type should be included.
